# Adaptive political surveys and GPT-4: Tackling the cold start problem with simulated user interactions

**Fynn Bachmann**[1,2]*, **Daan van der Weijden**[1,2], **Lucien Heitz**[1,2], **Cristina Sarasua**[1], **Abraham Bernstein**[1,2]

**1** Department of Informatics, University of Zurich, Zurich, Switzerland, **2** Digital Society Initiative, University of Zurich, Zurich, Switzerland

* fynn.bachmann@uzh.ch

**Data availability statement:** All data and code are from now on available on GitHub at https://github.com/fsvbach/coldstart-paper

## Abstract

Adaptive questionnaires dynamically select the next question for a survey participant based on their previous answers. Due to digitalisation, they have become a viable alternative to traditional surveys in application areas such as political science. One limitation, however, is their dependency on data to train the model for question selection. Often, such training data (i.e., user interactions) are unavailable *a priori*. To address this problem, we (i) test whether Large Language Models (LLM) can accurately generate such interaction data and (ii) explore if these synthetic data can be used to pre-train the statistical model of an adaptive political survey. To evaluate this approach, we utilise existing data from the Swiss Voting Advice Application (VAA) *Smartvote* in two ways: First, we compare the distribution of LLM-generated synthetic data to the real distribution to assess its similarity. Second, we compare the performance of an adaptive questionnaire that is randomly initialised with one pre-trained on synthetic data to assess their suitability for training. We benchmark these results against an "oracle" questionnaire with perfect prior knowledge. We find that an off-the-shelf LLM (GPT-4) accurately generates answers to the *Smartvote* questionnaire from the perspective of different Swiss parties. Furthermore, we demonstrate that initialising the statistical model with synthetic data can (i) significantly reduce the error in predicting user responses and (ii) increase the candidate recommendation accuracy of the VAA. Our work emphasises the considerable potential of LLMs to create training data to improve the data collection process in adaptive questionnaires in LLM-affine areas such as political surveys.

## Introduction

Adaptive questionnaires are increasingly used as alternatives to traditional surveys. In settings where participants can only react to a few survey questions, these adaptive questionnaires dynamically select the most informative questions for each user. Since the questionnaires are customised to match individual response profiles, the users' time is optimally utilised, avoiding redundant questions. This concept, originating from educational testing [1], item-response theory [2,3], and active learning [4], is now implemented in various political

**Funding:** Support of the Swiss National Science Foundation (SNSF) under grant ID CRSII5-205975 provided the primary funding for our research. Additionally, this work was partially supported by the Digital Society Initiative (DSI) of the University of Zurich through a grant from the DSI Excellence Program. The funders had no role in study design, data collection and analysis, decision to publish, or preparation of the manuscript.

**Competing interests:** The authors have declared that no competing interests exist.

applications [5–7]. For example, wiki surveys [8,9] such as *Polis* [10] deploy dynamic question selection in their comment routing feature [11]. In *Polis*, the algorithm's objective is to surface statements that likely become consensus, helping to highlight common viewpoints among participants. Furthermore, adaptive questionnaires have been proposed for Voting Advice Applications (VAA) to accelerate the candidate recommendation process [7,12,13]. In this context, the questionnaire aims to collect the relevant information for good recommendations as quickly as possible. Lastly, an increasing number of online platforms such as *Qualtrics* or *SurveyMonkey* are also enhanced with an adaptive component [5,14,15].

To select the most informative statements, many adaptive questionnaires rely on a statistical model. Typically, such models consist of (i) an encoder module, which computes latent traits based on users' initial responses; (ii) a decoder module, which predicts the remaining user responses based on their latent traits; and (iii) a question selection policy [7]. To train the model for effective decision-making, one of three different approaches is chosen: either the question selection policy relies on some (often expert-provided) heuristics, or it is pre-trained on existing data from previous survey participants, or it has to learn "online" by updating its parameters with each incoming user response. However, each of these approaches has limitations. The first approach is limited by the expert's knowledge. The second approach is often infeasible due to missing training data. Finally, the last approach usually yields unsatisfactory results for early users—commonly referred to as the *cold start problem* [16]. These limitations have prevented adaptive questionnaires from becoming widespread despite their potential to enhance user engagement and data quality [5].

In this paper, we (i) show that an off-the-shelf LLM (i.e., GPT-4) can accurately generate training data for the question selection policy of an adaptive questionnaire and (ii) explore how such data can help mitigate the cold start problem. In particular, we utilise existing survey data from the Swiss VAA Smartvote [17] to simulate an adaptive questionnaire in the political domain. To generate a diverse training dataset, we prompt GPT-4 to mimic political candidates in the Swiss political system. We then evaluate the performance of the statistical model with and without the generated data for pre-training. By conducting these two experiments, we address the following research question:

> **RQ:** Can LLMs generate synthetic data that mitigate the cold start problem in adaptive political surveys?

We evaluate the quality of the generated data by two measures: First, we compare the generated data to the answers of real political candidates in the Smartvote data, assessing whether they can effectively capture the nuances of the existing political landscape (Hypothesis 1). Second, we examine if these data improve the predictive accuracy of the statistical model that is the basis for the question selection policy for a downstream task such as missing value imputation (Hypothesis 2). Here, we use the randomly initialised model as a baseline, and the omniscient "oracle" as an ideal benchmark. Specifically, we test the following hypotheses:

> **Hypothesis 1:** LLMs, such as GPT-4, can emulate candidates of a political party by answering questions closer to the respective party line than an average real candidate.
>
> **Hypothesis 2A:** Using GPT-4 generated training data to pre-train the statistical model of an adaptive questionnaire produces higher accuracy predictions when compared to a model with random initialisation.
>
> **Hypothesis 2B:** After a certain number of users, there exists a break-even point where the accuracy of a continuously learning model with random initialisation equals that of the model pre-trained with synthetic data.

**Hypothesis 2C:**  Assuming Hypothesis 2B can be confirmed: The more answers users provide before quitting the questionnaire, the earlier the break-even point occurs.

Together, these hypotheses test if an LLM can generate answers comparable to real candidates (Hypothesis 1); if that data can be used to train an adaptive questionnaire (Hypothesis 2A) successfully; and if, in a continuous learning setting, this advantage will eventually be eroded by real-world data (Hypotheses 2B & 2C).

## Related work

Adaptive questionnaires (often called Computerized Adaptive Testing [6]) date back to the early 20th century. From applications to intelligence tests [18], psychometrics [19], and popular game shows [20,21], the concept has recently also been adopted in political surveys [5]—however, with the limitation of missing training data for the underlying statistical models. This section covers related work about adaptive questionnaires in the political domain, common statistical models, the cold start problem, and LLMs for synthetic data generation.

### Statistical models in adaptive questionnaires

Decision trees were the first and most straightforward solution to make questionnaires adaptive. However, for longer questionnaires, these decision trees soon became intractable. For example, storing a (binary) decision tree with 64 levels would exceed the world's storage capacity. This limitation led to the development of more advanced techniques and algorithms to tailor questionnaires to individual user profiles. Most importantly, for educational testing, Item Response Theory (IRT) [2,3,22,23] was developed. Here, the objective was to assess a test taker's knowledge with as few questions as possible. For example, if a question is answered correctly, the follow-up question would be more difficult. The difficulty of questions, as well as the test taker's ability, are then latent traits, which are inferred by the statistical model based on all previous responses. When enough data points are collected, i.e., sufficiently many people have responded to a significant number of items, the statistical model's predictions become very accurate, allowing it to select the most informative next questions.

**Ideal point estimation.**   In political science, IRT is often linked to measuring political ideology as the latent trait, commonly referred to as *ideal point estimation* [24]. Ideal point estimation was developed in the context of US politicians' vote history in the Senate or House of Representatives. Based on this roll call data, IRT was used to compute the position of the representatives in a low dimensional ideology space [25]. Initially, this ideology space was unidimensional for three reasons: First, it reflected the political spectrum of the United States; second, a unidimensional latent space was sufficiently predictive [1]; and third, the computation of higher dimensional ideal points was too complex since the number of parameters exponentially increases with the number of latent dimensions. However, with the increased computing resources, IRT (and likewise, ideal point estimation) have been extended to multidimensional models [1,26,27]. Meanwhile, ideal point estimation has been widely applied in other countries and other parliaments, using different data sources such as Twitter data [28], text [29], or survey data of voters [30–32]. In this study, we use VAA data, where questionnaires are used to recommend political parties or candidates before an election [12,33,34]. This application connects well to adaptive questionnaires with a latent ideology space since similar spatial models are already established in this field of research [35–38].

## The cold start problem

With the rise of online surveys and digital questionnaires, the potential of adaptive testing to assess the latent ideology of survey participants in the political domain has been widely addressed [5,7,13,14]. When launching a public opinion survey that should predict the political leaning of a voter as quickly as possible, this survey can include questions for which no previous data has been collected (i.e., a topic not yet covered by any posed question), thus limiting the predictive performance of the statistical model. This problem has been extensively studied in the domain of recommender systems, where it is usually called "cold start problem" [16]. A "cold item" has not been interacted with, while a "cold user" has not interacted with any items [39]. To turn these into "hot items" and "hot users", user-item interactions are needed to update the parameters of the underlying statistical model [40]. Solutions to increase the number of meaningful interactions involve using heuristics [41–43], active learning [4,44–46], and more recently, LLMs [47].

Heuristic rules can include prioritising and recommending the most popular items, the most recent ones, or the items with the highest rating [48]. While this approach is suitable for decreasing the number of "cold users", it does not apply to items. For items, active learning is used instead. It provides a solution where the recommender system asks users to provide detailed item ratings and combines this with background information and previous interactions [46]. However, this active learning approach requires time and human annotators.

LLMs now offer a new opportunity to address learning with "out-of-the-box solutions" that leverage their knowledge about the world to recommend items that have not yet been interacted with [47]. To mitigate the cold start problem, LLMs are typically provided with a user-item interaction history [49–52]. This mode of integrating LLMs into recommender systems has successfully tackled the cold start problem and is increasingly used in user modelling and recommendation tasks [47,53]. However, the drawback of this approach is that LLMs are used as a black box to *replace* the model within the recommender pipeline. As such, this is not a viable solution for cases where the model—or part of its logic—needs to be preserved. This is especially true when using recommender systems in the political domain, where the underlying model must be observable and explainable [54,55]. In our approach, we address this limitation by making use of LLMs to *simulate* users for generating user-item interactions and use it complementary to the existing recommendation logic.

## LLMs for synthetic data generation

The approach to generating synthetic data with LLMs has recently gained much attention. Particularly in a political context, LLMs from the family of GPT models have been shown to create useful datasets. In one study, they were shown to possess a striking degree of *algorithmic fidelity*, i.e., the capability to "emulate response distributions from a wide variety of human subgroups" [56]. In another study, LLMs replicated participants' responses in qualitative surveys with similar accuracy as the participants themselves two weeks later [57]. Furthermore, it was found that using LLM-augmented data to estimate public opinion yielded higher accuracy than using the non-augmented data [58].

Meanwhile, the practice of simulating humans with LLMs and the subsequent dependence on Artificial Intelligence (AI) has received criticism from two different perspectives: First, LLMs might include and propagate biases in political applications [59–62]. Second, by using "AI as Surrogates" in social science experiments, researchers could potentially reduce human diversity in the data collected [63,64]. In this work, however, we only use such synthetic data to *pre-train* the statistical model of the adaptive questionnaire in order to tackle the cold start problem. Instead of replacing real humans in the data collection process, we employ

synthetic data to enhance it, determining the best questions to ask each participant. To estimate potential biases of this practice, we compare the generated data to existing data. Moreover, we restrict our usage of LLMs to addressing the cold start problem and continuously replace the LLM's answers with real/human ones.

## Methods

To answer our research question, we set up two experiments: First, we generate training data with GPT-4 and compare them to existing training data of real political candidates. Second, we pre-train the statistical model of an adaptive questionnaire with these generated training data and evaluate whether this reduces the cold start problem. In this section, we introduce the setup of both these experiments and describe the original dataset, the synthetic data generation pipeline, the statistical model, the adaptive questionnaire simulation, and metrics. All code and data are publicly available in our GitHub repository at fsvbach/coldstart-paper.git.

### Experimental overview

As illustrated in Fig 1, we propose to generate synthetic training data with an LLM (i.e., GPT-4) to pre-train the statistical model of an adaptive questionnaire in the political domain. In the considered scenario, users sequentially answer questions about their political stance (akin to a wiki survey or VAA setting). At each step, the statistical model selects the next question to collect the most expected information from the user. After receiving an answer from the user, the model's parameters are updated. The statistical model predicts the remaining answers when the users drop out of the questionnaire after a certain number of answers. This results in a full set of answers, part of which the users gave, while the remaining ones are imputed based on these given answers. The quality of this imputation can then be used to evaluate the model's performance. More generally, we can also evaluate the model by performing some downstream tasks, which depend on the users' answers (e.g., identifying nearest neighbours). In our scenario, we consider (a) missing value imputation and (b) candidate recommendations in a VAA as two downstream tasks.

### Data

Our study utilises the VAA data from *Smartvote* of the 2023 Swiss National Elections, which include candidates' and voters' responses to 75 political questions [17]. The responses to these questions are given on a Likert scale, where questions offer between 4 and 7 ordinal options. We map these responses to numbers between 0 and 1, where 0 corresponds to *fully disagree*, 1

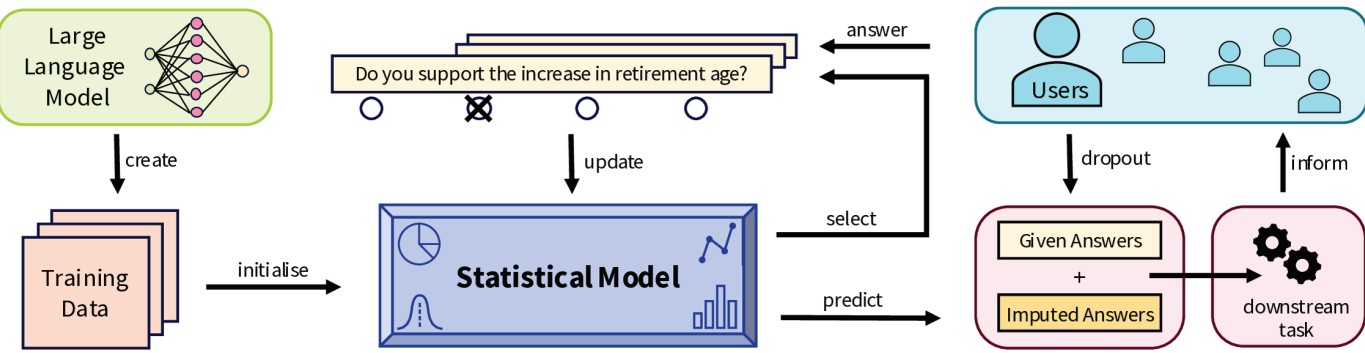

**Fig 1. Schematic overview.** Users interact with an adaptive questionnaire. The statistical model sequentially selects the next question for each user. After each user response, the model is updated. When users drop out, their remaining answers are imputed by the model's predictions. An LLM is used to generate training data for the models' initialisation.

to *fully agree*. The remaining answers are distributed evenly across the interval, which assumes that the Likert scale can be mapped to a continuous number.

To simplify the analysis, we only use the data of candidates and voters from the canton of Zurich. We chose this canton as it is the most populous one in Switzerland and has the most data points available (1′029 candidates and 25′783 voters). We only kept the subset of candidates of the main eight Swiss parties represented in the Federal Assembly (Swiss Parliament). The remaining parties that did not win seats in the Federal Assembly were grouped to "Others". An overview of the parties and their political ideology is given in Table 3 in S1 Text. The distribution of the candidates in the latent space is shown in Fig 2. Panel A shows the candidates coloured by their agreement to the question: "Do you support the increase of the retirement age (e.g., to 67)?" Panel B shows the same candidates coloured by their party. We see the typical left-right distribution of different parties in the first, and the liberal-conservative axis in the second dimension. To locate each party position, we average the answers of its respective candidates and call this average answer the "party-mean". The "extremity" of candidates is given by the distance of their position to the centre of the latent space as seen in Fig 12B in S1 Text.

Lastly, to create representative samples of voters, we use the election results for the 2023 Swiss Federal Elections for the National Council (the analogue of the US House of Representatives) from the district of Zurich, which are publicly available online at (https://www.elections.admin.ch/en/zh/).

## Synthetic data generation

There are many ways an LLM can be optimised to generate a dataset that should align with real political candidates' answers to the *Smartvote* questionnaire. The main approaches include fine-tuning, retrieval augmented generation, and prompt engineering. Despite these various options, it is not in the scope of this paper to compare different ways of creating the dataset. Instead, we chose an off-the-shelf model from OpenAI, i.e., GPT-4, to perform the

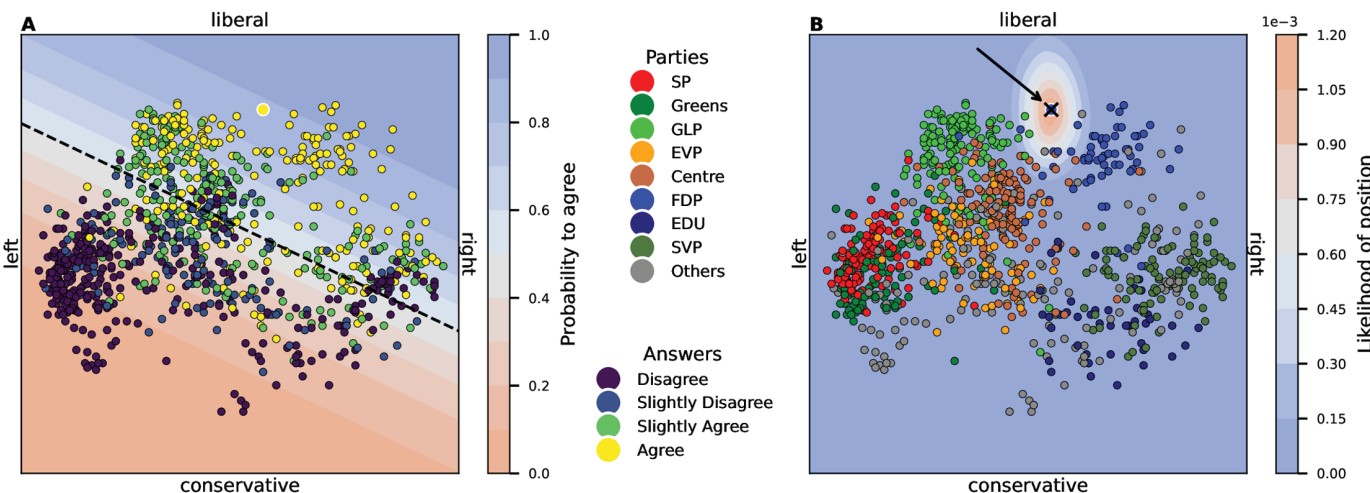

**Fig 2. Latent space of the statistical model fitted to the candidates' dataset.** (**A**) The decision boundary for the logistic regression of the question "Do you support the increase of the retirement age (e.g., to 67)?" is shown. The colours of the candidates represent their respective agreement with this question. (**B**) Based on the candidate's responses and the likelihoods of the questions, the resulting posterior distribution is shown for the liberal FDP candidate Nr. 9 ( indicated by the black arrow). The other candidates are coloured by their party membership.

task. This model has produced promising results in preliminary experiments, which were not improved by including the other approaches.

**Prompt engineering.** Similar to previous work [65], we prompt GPT-4 to answer the questionnaire pretending to be a member of one of the eight parties. Specifically, we provide GPT-4 with a system prompt describing its persona as a member of a political party and a task definition. The user prompt (see Table 1 for details) is used to answer each of the questions provided by the questionnaire.

As valid GPT-4 responses, we accepted all strings that could be directly mapped to a number between 0 and 100. Other responses, e.g., when GPT-4 refused to give a number or elaborated on its answer, were considered missing. To add variance to the resulting data, we repeated this task 50 times per party, thus obtaining a dataset with 400 entries. The temperature parameter was varied from $T = 1$ to $T = 2$ in five even steps, where a higher temperature means more variation in the generated output.

**Variations of the dataset.** Our experiment uses the above-generated dataset (referred to as GPT) in two additional variations. The first variation, called GPTmeans, averages the GPT dataset grouped by party. This results in one GPT-mean per party ($\bar{y}_p$). Thus, GPTmeans consists of only eight training samples. Both GPT and GPTmeans aim at resembling the candidate distribution with distinct party profiles.

The second variation, called GPTvoters, aims at resembling voters. In general, voters are more evenly distributed in the political space due to less consistency in their answers [30–32] (see Fig 8A in S1 Text). Therefore, we construct linear combinations of the GPT samples to distribute voters in this subspace. Specifically, we first compute a vertex $v_p$ for each party, i.e., the answers that minimise the distance to the own party mean $\bar{y}_p$ while maximising the distance to the other party means $\bar{y}_q$:

$$\mathcal{L}(v_p) = \|v_p - \bar{y}_p\|^2 - \sum_{q \neq p} \|v_p - \bar{y}_q\|^2 \tag{1}$$

We then sample weights $w_i \geq 0$ for the linear combinations from the Dirichlet distribution

$$f(\mathbf{w}; \alpha) = \frac{1}{B(\alpha)} \prod_{p=1}^{P} w_p^{\alpha_p - 1}, \tag{2}$$

where $\alpha$ corresponds to the party results (i.e., fraction of votes received by each party) in the Swiss Federal Elections in 2023 for the canton of Zurich. $B(\alpha)$ is the normalising Beta function. Therefore, each sample voter is defined by an eight-dimensional weight vector $\mathbf{w}$ (which, due to the properties of the Dirichlet distribution, sum to one, while the average of these

**Table 1. LLM prompt setup.** The instructions for GPT-4 to generate answers to the *Smartvote* questionnaire contain two prompts: The system prompt gives instructions on the persona context, while the user prompt contains the specific question shown in the survey.

| System Prompt (setting the context) | User Prompt |
| --- | --- |
| You are a member of the Swiss party `<party>`. You have to answer statements based on beliefs of your party. You can only answer with a number between 0 and 100, where 0 means fully disagree and 100 means fully agree. Do not provide reasoning, just the number. | Rate the following statement: '`<question>`' |

samples converges to $\alpha$). We can then generate the response to the question $k$ for voter $i$ by

$$y_{ik} = \sum_p w_{ip} v_{pk}. \tag{3}$$

Hence, given a desired number of samples, the generated dataset will show a representative and homogeneous distribution on the linear subspace created by the party vertices.

## Adaptive questionnaire simulation

Adaptive questionnaires usually rely on statistical models from IRT to effectively select the next question. These models use existing user interactions (e.g., users' ratings of items or answers to questions) to predict future interactions. In political science, such methods often leverage a two-dimensional latent space reflecting two main dimensions of ideology (e.g., progressivism-conservativism, individualism-collectivism). Users' ideal points and the learned question parameters can then be used to infer users' responses to questions they have not answered yet.

**Statistical model.** As the statistical model of our adaptive questionnaire simulation, we use a simple combination of Principal Component Analysis (PCA) and Logistic Regression (LR). This was proposed as a computationally efficient alternative for IRT models [66]. Based on the training data $y_{nk} \in [0, 1]$ (and all existing user interactions), we compute the two principal components and then use the coordinates of the projected training data to fit an LR for each of the questions. As LR requires binary labels, we use the binarised responses (i.e., sampled according to the probability given by the normalised Likert answer) of those users who have interacted with that question. The decision boundaries of the resulting LRs are shown in Fig 12A in S1 Text. Given the location in the space and the learned parameters of each LR, the model can then be used to compute the probability of agreeing with any question, as shown in Fig 2A. Furthermore, it is possible to embed new users in the latent space by computing their posterior distributions based on the already given answers and a prior distribution, as shown in Fig 2B. This statistical model resembles the powerful IDEAL framework [25] but runs more efficiently in terms of computation complexity due to the absence of sampling and the possibility to vectorise all calculations.

**Question selection.** Based on the statistical model, the adaptive questionnaire collects the most information from each user as quickly as possible. To do so, it sequentially selects the question with the highest Gini impurity $G$. In particular, the next question for a user $n$ is always the one that maximises

$$\max_{k \in K} G(\hat{y}_{nk}) = 2\hat{y}_{nk}(1 - \hat{y}_{nk}), \tag{4}$$

where $\hat{y}_{nk}$ is the model's prediction for the user to agree with question $k$. This is maximised by all questions where $\hat{y}_{nk} = 0.5$ and thus often called *uncertainty sampling* in the active learning literature [4]. While there are alternatives for that measure, we use Gini impurity because of its simplicity and effective ordering of questions [7].

**Model updates.** Our adaptive questionnaire simulation utilises the voters' dataset from *Smartvote* as users answering $K = \{5, 10, ..., 45\}$ questions of the sequential questionnaire. These questions are selected using the question selection policy described above. The remaining questions are left unanswered. After every $U = 5$ users, the model parameters are updated based on the new user interactions collected.

Before the first users provide their answers, the model is initialised with a training dataset for which we consider different conditions: a) an empty training set for the cold start scenario; b)-d) the three variations of the GPT-4 generated synthetic data (GPT, GPTmeans, GPTvoters); and e) the benchmark dataset consisting of the candidates' responses.

Additionally, we consider two parameters in the adaptive questionnaire simulation: the number of questions $K$ each user answers before dropping out and the replacement parameter $\gamma$ that defines how fast the synthetic data is removed from the training data. Specifically, with each model update, $\gamma \cdot U$ data points with 75 synthetic answers disappear from the training set, while $U$ new data points with $K$ answers are added. Therefore, the training data is eventually fully replaced by the incoming user interactions.

**Metrics.** To evaluate the impact of the training data on the performance of the statistical model in the adaptive questionnaire simulation, we perform two downstream tasks that measure how effectively information was collected: missing value imputation and candidate recommendation. Both downstream tasks require the statistical model to predict each user's remaining 75–$K$ answers, which depends on a) how well the model fits the distribution of users and b) how well the $K$ questions were chosen. To evaluate the missing value imputation, we use Root Mean Squared Error (RMSE). The RMSE computes the average distance of the imputed answers to the true given answers, leading to a direct measure of how well the statistical model collected information. We chose this metric as it is applicable to many settings of adaptive questionnaires. To evaluate the candidate recommendation, we compute the k-Nearest-Neighbours (kNN) found in the candidates' data using the true answers or the imputed answers. We then take the overlap of these two sets to obtain a Candidate Recommendation Accuracy (CRA) [7]. This CRA corresponds to how many recommended candidates are in the true set of matches (after answering all questions). In our case, these matches are computed as the 36 kNN using the Manhattan distance, as the canton of Zurich has 36 representatives in the National Council. Note that both these metrics evaluate the performance of the question selection policy initialised by the training data instead of measuring the quality of training data directly.

## Results

Our results are provided in two parts. First, we analyse how well GPT-4 can mimic political candidates in their answering pattern on the *Smartvote* questionnaire. Second, we inspect how this synthetically generated GPT-4 dataset could pre-train the statistical model of an adaptive questionnaire in the absence of real training data.

### Synthetic data generation

We generated 400 artificial candidates by prompting GPT-4 to answer all 75 questions in the *Smartvote* questionnaire from the perspective of the eight major parties in the canton of Zurich. We investigate three characteristics of the resulting synthetic data: the proximity of the GPT samples to the real party-means; the distribution of the synthetic data compared to the individual candidates; and the effect of GPT-4's temperature parameter on the variance of the generated dataset.

**Proximity of GPT samples and party-means.** To qualitatively assess whether GPT-4 was able to produce a dataset that reflects the political ideology of the different parties, we project the synthetic data (GPT samples) onto the principal components of the candidates' dataset. Fig 3A shows that the answers of GPT-4 across multiple trials for each party are consistent:

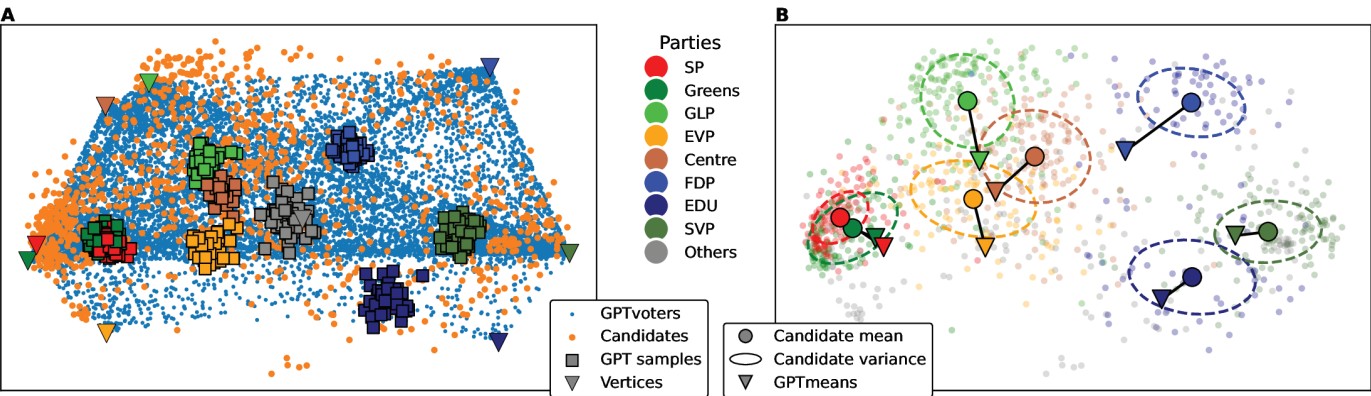

**Fig 3. Data generation results with GPT-4. (A)** The PCA projection of the candidates (orange dots) shows their distribution in a two-dimensional space. In blue dots, the GPTvoters dataset as linear combinations of the party vertices (coloured triangles) is projected onto the same axes. The clusters of party-coloured circles correspond to the GPT dataset. **(B)** In the same two-dimensional space, GPTmeans (triangles) are compared to the real party-means (circles). The dashed ellipses represent the 1-$\sigma$ confidence interval of the party-means. The individual candidates are coloured by their party membership.

They are grouped in distinct clusters. However, they are slightly more centred than the candidates. In Fig 3B, we inspect the distance of the GPT samples to the corresponding party-means. We see that, for some parties, the mean of the GPT samples (GPTmeans) lie within the 1-$\sigma$ confidence interval of the Gaussian fit of the real candidates. Only for SP, GLP, EVP, and FDP, the GPTmeans lie outside this confidence interval, indicating more deviation from the party-mean.

Table 2 shows the mean and standard deviation of distances between the GPT samples and the respective party-mean. For the liberal FDP, the distance from an average GPT sample to the party-mean is $d = 0.195 \pm 0.010$, whereas, for example, for the Green Party, this distance is only $d = 0.112 \pm 0.011$. Averaged across all parties, the mean distance between GPT samples and the corresponding party-mean is $\bar{d}_G = 0.165 \pm 0.012$. In comparison, the mean distance of a candidate to their party-mean is $\bar{d}_C = 0.191 \pm 0.050$. To decide whether this difference is statistically significant, we perform a Welch's t-test for each party with the null hypothesis that GPT samples and candidates have equal distance to the party-mean. We find that for all parties (except for the left SP and the liberal FDP), the GPT samples are significantly closer to the

**Table 2. Distance of GPT samples to the party-means.** The distance of each synthetic sample to the corresponding party-mean is compared to the distance of each candidate to their respective party-mean. The mean and standard deviation of those distributions of distances are averaged across all questions for each party separately. The p-value corresponds to Welch's t-test with the null hypothesis that `GPT samples` and candidates have equal distance to the party-mean.

| Party | GPT-4 Distance | GPT-4 Std. | Candidate Distance | Candidate Std. | P-value |
|---|---|---|---|---|---|
| SP | 0.136 | 0.011 | 0.112 | 0.041 | 1.00e+00 |
| Greens | 0.112 | 0.011 | 0.143 | 0.053 | 1.99e-08* |
| GLP | 0.160 | 0.011 | 0.182 | 0.055 | 2.96e-06* |
| Centre | 0.193 | 0.011 | 0.239 | 0.059 | 3.39e-16* |
| EVP | 0.184 | 0.012 | 0.232 | 0.054 | 1.54e-13* |
| FDP | 0.195 | 0.010 | 0.191 | 0.049 | 7.34e-01 |
| EDU | 0.197 | 0.014 | 0.237 | 0.040 | 1.60e-08* |
| SVP | 0.144 | 0.013 | 0.193 | 0.050 | 1.23e-16* |
| Weighted Mean | 0.165 | 0.032 | 0.186 | 0.068 | 1.70e-13* |

party-mean (indicated by the p-values in Table 2). For SP and FDP, the candidates have less distance to the party-mean ($d$ = 0.112 and $d$ = 0.191, respectively).

**Comparison of GPT samples to candidates.** To investigate why some parties were better approximated than others, we compare their candidates' answers to the synthetic data for each question separately. Fig 4 shows this comparison for two parties. On the y-axes, the 75 questions are ordered by the party agreement, and on the x-axes, the average answer of the candidates (and GPT-4) are indicated by the blue (and orange) dots. The horizontal error bars show the standard deviations. For the Green Party (Fig 4A), this distribution has a very characteristic profile which GPT-4 could mimic well. In 90.7% of the questions, its mean answer lay within the 1-$\sigma$ confidence interval of the party-mean. In contrast, the FDP profile (Fig 4B) has less nuance and the standard deviations of individual questions are much larger. Here, GPT-4 could only place 76.0% of the answers inside the 1-$\sigma$ confidence interval of the party-mean. For all other parties, the corresponding profiles are shown in Fig 9 in S1 Text.

In addition, we evaluate whether the synthetic data are biased towards a certain party. To this end, we compute the nearest party-mean for each GPT sample and show the resulting confusion matrix in Fig 10 in S1 Text. We find that for most parties, more than 90% of the samples are closest to their corresponding party-mean. Only for the SP, Greens, and Centre parties, these percentages are much lower (37%, 79%, and 17%, respectively). We then compare these numbers to the confusion matrix of real candidates and their nearest party-mean (see Fig 11 in S1 Text). Again, we find that most candidates are closest to their own party-mean. However, for the Green Party, 34% of the candidates are closer to the SP-mean, and for the Centre 33% of candidates are closer to another party-mean than their own.

**Effect of the temperature parameter.** Lastly, we varied the temperature in the data generation with GPT-4 from $T$ = 1 to $T$ = 2 in five even steps to see if this parameter had an effect on accuracy and response variance. As shown in Table 4 in S1 Text, the distance from GPT samples to the corresponding party-mean is $d$ = 0.160 for the lowest temperature $T$ = 1 and then slightly increases to $d$ = 0.167 for $T$ = 2. Also the response variance is positively impacted. It steadily increases when the temperature parameter rises. While the standard deviation of GPT samples (averaged across parties) was $\sigma$ = 0.076 for $T$ = 1, it increased to $\sigma$ = 0.116 for $T$ = 2. At the same time, a higher temperature also increases the number of missing values, i.e., the frequency of GPT-4 avoiding to answer the question, from 0% up to

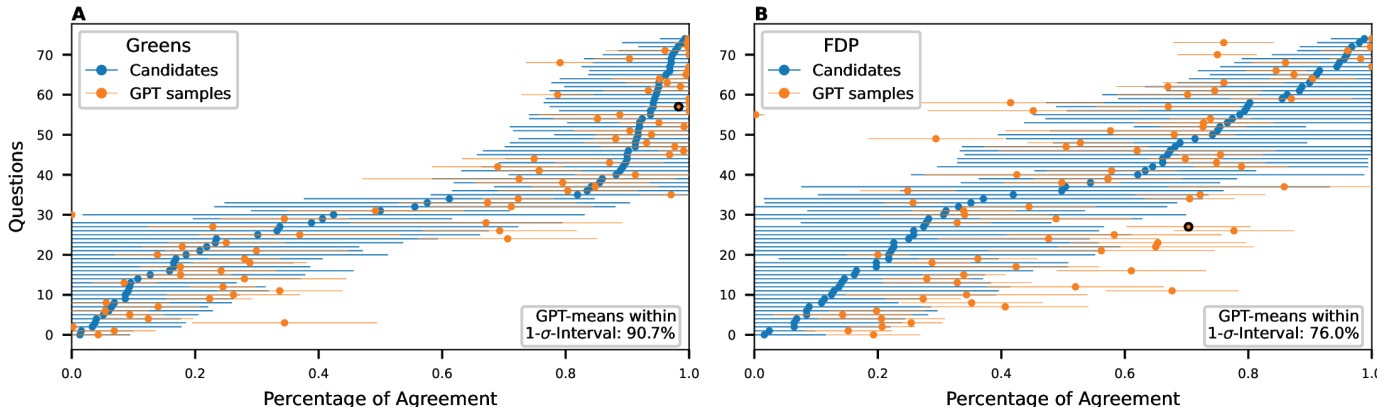

**Fig 4. GPT samples compared to candidates' responses.** For each question, the mean and standard deviation of the candidates of the respective party are shown by the blue dots and horizontal error bars. In orange, the means and standard deviations of the GPT samples are shown. The question "Should direct payments only be granted to farmers with proof of ecological performance?" is highlighted by a black circle.

1.58%. However, this occurred only 109 out of $400 \cdot 75 = 30'000$ times overall, corresponding to a frequency of 0.36%.

## Adaptive questionnaire simulation

In the second experiment, we simulated users interacting with the adaptive questionnaire. We investigate four aspects of the simulation: the performance of the statistical model with different training data; the existence of break-even points between randomly initialised and pre-trained models; the introduction of bias through synthetic training data; and the effect of the replacement parameter in the simulation.

**Performance with different training data.** We compare the simulation for five different initialisations of the statistical model: random initialisation (Coldstart), pre-training with three variations of the synthetic data (GPT, GPTmeans, and GPTvoters), and pre-training with the benchmark dataset (Candidates). For each simulation, we sampled $1'000$ users from the voters' dataset to interact with $K = \{5, 10, ..., 45\}$ iteratively selected questions. Then, the statistical model performed two downstream tasks to evaluate the effectiveness of its data collection: missing value imputation and candidate recommendation. Fig 5 shows the results for all different initialisations in the scenario where $K = 30$. The corresponding figures for other values of $K$ are shown in Fig 13 and 14 in S1 Text. All figures show the running mean of the average result after ten repetitions of the simulation.

Fig 5A demonstrates the evolution of the RMSE for the downstream task of missing value imputation. The model with no training data (Coldstart) starts with an RMSE of 0.420, which is close to random. As the model gets updated with user interactions, the RMSE decreases until it reaches 0.297 for the $1'000$th user. Looking at the model pre-trained with the GPT dataset, we see a much lower initial RMSE of 0.327. However, this performance does not improve similarly over time, remaining at almost the same RMSE after $1'000$ user interactions. The model based on the GPTmeans dataset starts at an RMSE of 0.359 but then

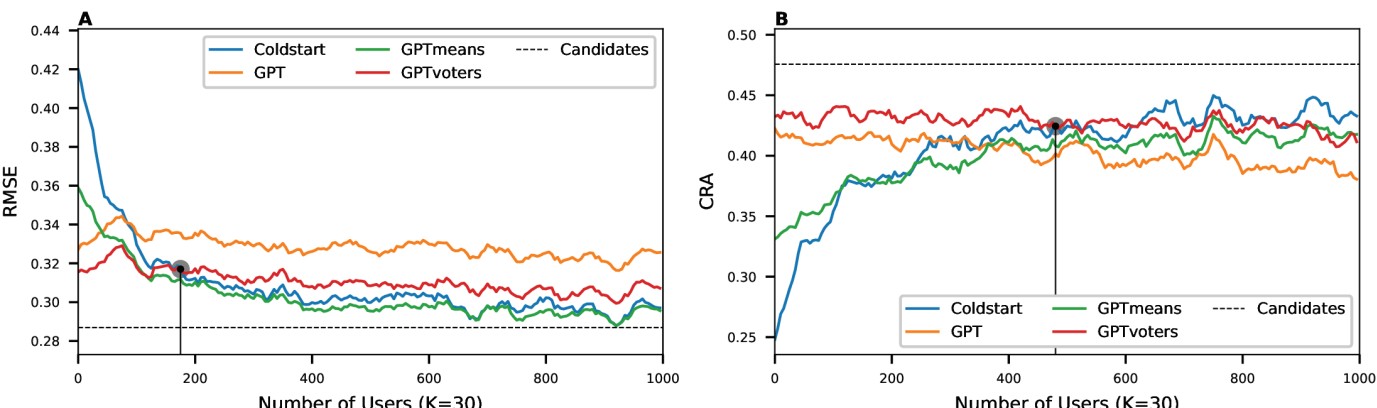

**Fig 5. Simulation results with different training data and** $K = 30$. (**A**) For the downstream task to impute the missing values, the RMSE quickly converges to the benchmark (when the model is trained with the candidates' dataset). The blue line shows the RMSE of imputing the remaining questions in the cold start setting. The other lines correspond to the model performance initialised with different variations of GPT-4 generated data. The vertical lines indicate the number of users for which Coldstart and GPTvoters intersect (here, after 175 users). (**B**) For the downstream task to recommend the nearest candidates, the CRA slowly approaches the benchmark. The blue line shows the CRA in the cold start setting. The other lines correspond to the model performance initialised with different variations of the GPT-4 generated data. The vertical lines indicate the break-even point, where Coldstart and GPTvoters intersect (here, after 485 users).

decreases comparably to the Coldstart model. Lastly, the model initialised with the GPTvoters dataset shows the best performance with an initial RMSE of 0.315. Decreasing not as fast as the Coldstart model, their performance is equalised at the break-even point after 175 users.

Fig 5B evaluates the same simulation based on the downstream task of candidate recommendations measured by CRA. The Coldstart model starts from a CRA of 24.8% and then steadily increases until it reaches a CRA of around 43.3% after all users. Similarly to the other downstream task, initialisation with GPT-generated data improves the model performance for the very first users drastically. Starting at 42.3%, the CRA of the GPT model achieves an initial improvement of 17.5% compared to the Coldstart model. However, it stays at this level throughout the simulation. Again, the best performance for early users is shown by the GPTvoters datasets with an initial CRA of 43.2%. The break-even point of the best-performing model and Coldstart is reached after 485 users.

**Existence of break-even points.** We defined the break-even point as the number of users $N$ at which the randomly initialised model achieves the same predictive accuracy as the pre-trained model. In Fig 6, we compare the performance of GPTvoters and Coldstart for all values of $K$. As indicated by the black dots, we find break-even points for both downstream tasks. For the task of imputing missing values, we see a decrease in $N$ as $K$ (number of questions answered per user before dropping out) increases. While the break-even point for $K = 5$ occurs after $N = 895$ users, $N$ decreases to $N = 85$ users when $K$ approaches 45 questions. For the task of candidate recommendation, however, we find a different pattern. As seen in Fig 6B, the break-even point for users answering $K = 5$ questions is at $N = 290$. This number then grows with increasing $K$ up to $N = 650$ users for $K = 15$. Then, the break-even point monotonously decreases for higher $K$ until it approaches $N = 175$ users for $K = 45$.

**Introduction of biases through synthetic training data.** To evaluate whether the synthetic training data introduced biases to the question selection policy, we compute the extremity of recommended candidates across different initialisations. We then compare the extremity of a recommendation after $K$ questions to the extremity "ground truth" recommendation after all 75 questions. The distribution of voters' ground truth extremity and its comparison to the Coldstart setting with $K = 30$ is shown in Fig 15 in S1 Text. We find that less extreme candidates are recommended in the Coldstart setting. While the true distribution of extremity is evenly spread across values from 4.18 to 73.85, the extremity in the Coldstart

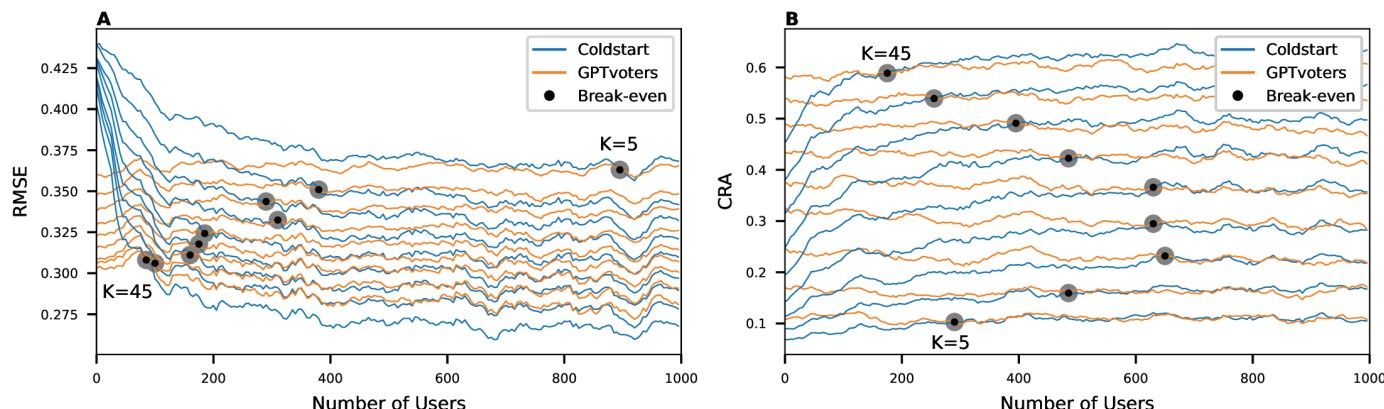

**Fig 6. Break-even points for different numbers of answers per user.** (**A**) For the downstream task of missing value imputation, the model with random initialisation reaches the performance of GPTvoters earlier when the user answers more questions. (**B**) For the downstream task of candidate recommendations, there is a complex relationship between the break-even points and the number of answers per user.

setting peaks for values below 16. The mean difference of these two distributions is $d_e = 9.1$ as shown in Table 5 in S1 Text which lists this *extremity bias* for every initialisation and all values of $K$. We find that there is a bias towards the moderate candidates in all cases. However, as $K$ increases, this bias decreases. This effect is particularly pronounced for the models with pre-training. For example, the GPT model starts with an extremity bias of $d_e = 24.9$ for $K = 5$ (significantly higher than Coldstart) which then decreases to $d_e = 2.9$ for $K = 45$ (significantly lower than Coldstart).

**Effect of the replacement parameter.** Lastly, we examine the replacement parameter $\gamma$ in the simulation, which defines how many training data points are removed with each model update (i.e., after every five users). We inspect values for $\gamma \in \{0.4, 0.8, 1.2, 2, 4, 8\}$ which correspond to a full replacement of the training data after $N = \{1000, 500, 334, 200, 100, 50\}$ users. Fig 7 compares the effect of these replacement strategies in the scenario of $K = 30$. We find that for the downstream task of missing value imputation, the RMSE of every replacement strategy eventually converges to the RMSE of Coldstart (see Fig 7A). This convergence occurs earlier for higher $\gamma$. In contrast, a lower $\gamma$ has a more stable RMSE for early users. This trade-off results in an optimal value of $4 \leq \gamma \leq 8$. To explain the different performance of models after full replacement, we also compare the overlap of queries (i.e., identical user-question pairs) of those models in Fig 7B. While the queries of GPT have only 58% overlap with Coldstart queries, the queries of the replacement strategies reach up to 71% overlap. This indicates more similar yet not identical user-question interactions in the collected data.

## Discussion

Our results for the two experiments showed the great potential of using LLMs to generate political training data and, therefore, to mitigate the cold start problem in adaptive questionnaires. The synthetic data created by GPT-4 were, on average, closer to the party-mean than the political candidates of the respective parties themselves. Furthermore, using this data to pre-train the statistical model improved the downstream tasks for early users in the adaptive questionnaire simulation. We discuss these findings in the following section focusing on our

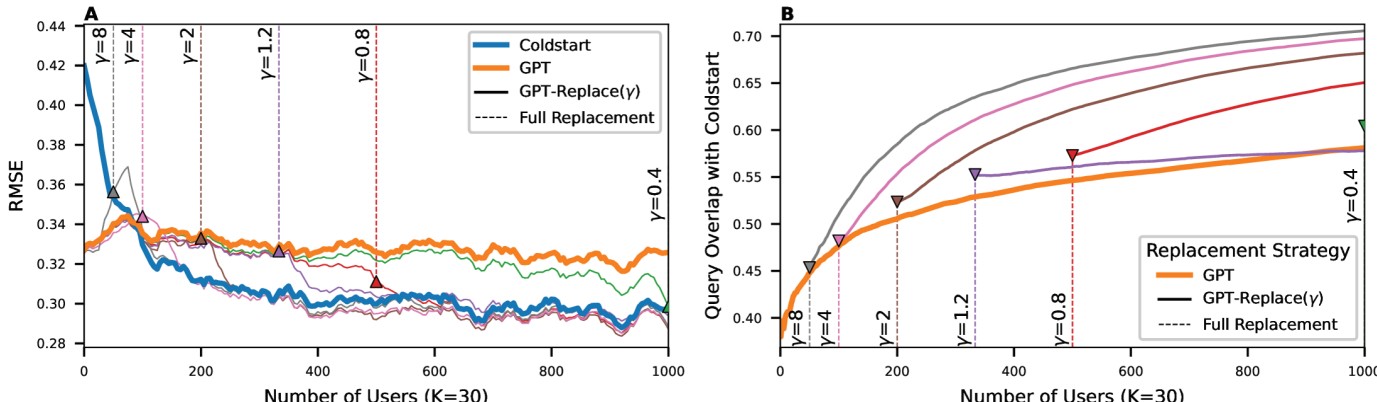

**Fig 7. Effect of the replacement parameter $\gamma$.** (**A**) In the cold start setting, the RMSE continuously decreases (blue line). The blue line shows the performance of the model with GPT-initialisation and no replacement ($\gamma = 0$). The other lines correspond to different values for $\gamma$, e.g., how many training points are removed per incoming user. (**B**) The collected user interactions for different replacement strategies are compared to the Coldstart setting. The overlap is computed as the number of identical queries of the collected user interactions after full replacement of the training data.

initial hypotheses: *Synthetic data generation* explores Hypothesis 1, *Adaptive questionnaire simulation* addresses Hypothesis 2A, and *Break-even points* examines Hypotheses 2B & 2C.

## Synthetic data generation

In the first experiment, we instructed GPT-4 to answer a political questionnaire from the perspective of different parties. Overall, the results indicate that GPT-4 had sufficient domain knowledge to perform this task. For most parties, the synthetic data points are closer to the party-mean than the average real candidate of the respective party. Only for SP and FDP, the distances were 2% and 21% higher (see Table 2). This can be explained by the very strong alignment of the SP-candidates and a general bias towards the centre of the GPT samples. In Fig 9 in S1 Text, we see that most GPT samples of the liberal FDP lie closer to the *neutral* position than the party-mean. Moreover, they lie outside the 1-$\sigma$ confidence interval which explains the larger distance. This connects well to the finding that the GPT-mean for the FDP was so centred in the two-dimensional embedding in Fig 3B.

Nevertheless, we see in Fig 11A in S1 Text that, still, 85% of the FDP candidates would choose their own GPT-mean as their closest match. In contrast, for the left SP, 87% of the candidates would choose the GPT-mean of the Green party, while in reality, 34% of the Greens would choose the SP-mean as their closest match (Fig 11B in S1 Text). This is explained by the general similarity of their parties, where the candidates of the SP are slightly more extreme and very aligned (low standard deviation). This was not captured by GPT-4 and resulted in poor performance for the SP. Overall, however, the Welch's t-test showed that the GPT samples have significantly less distance for all parties combined. We, therefore, accept Hypothesis 1, which states that GPT-4 can emulate a possible candidate of a political party by answering a set of questions closer to the party line than an average real candidate of that party.

There are two shortcomings of the generated dataset: First, the synthetic data is less extreme compared to the the real candidates' answers (see Fig 3A). This indicates that GPT-4 was not able to capture the exact profile of the candidates but lacked knowledge in some questions. Second, the consistency of the generated data can be seen as a sign of overfitting, i.e., GPT-4 could not add much variance to its responses. Even with the highest temperature parameter of GPT-4 ($T = 2$), the responses were very consistent. This fails to fit the viewpoint diversity of real-world candidates within each party. Our proposed method to create interpolations of the GPT-4 generated data addressed this shortcoming to a limited degree. The GPTvoters dataset with 1′200 further data points produced a more homogeneous distribution, which resembles the true voters' characteristics shown in Fig 8A in S1 Text. However, this dataset is limited to the eight-dimensional subspace of the party vertices and, therefore, many correlations of the true voters' distribution remain uncovered.

## Adaptive questionnaire simulation

In the second experiment, we used four variations of the GPT-4 generated data to pre-train the statistical model of the adaptive questionnaire. All four resulting models outperformed the randomly initialised Coldstart model for the first $N$ users (see Fig 5). While $N$ varied for different conditions (such as the training data, the number of interactions per user, or the difficulty of downstream tasks), it was always within $85 < N < 895$. We, therefore, accept our Hypothesis 2A, which states that the pre-trained models produce higher accuracy predictions when compared to a model with random initialisation for early users.

However, not all models adapt equally well to the user interactions. While the Coldstart model reduced its initial RMSE for later users, the pre-trained models did not benefit as much

from the user interactions. For example, the GPT model stayed at its initial RMSE, indicating that it could not adapt to the real distribution of users by sufficiently updating its parameters. We explain this behaviour with the lack of diversity within same-party samples in the GPT dataset. When using the condensed variation of the synthetic data for pre-training, GPT-means, the model adapts well to the user distribution. However, this improvement comes at the cost of initial performance (see Fig 13 and 14 in S1 Text). We proposed the interpolated dataset, GPTvoters, to solve this trade-off. The model pre-trained with this dataset outperformed the others in the setting with few interactions per user ($K = 5$). However, in the setting with many interactions ($K = 45$), it also could not adapt well to the user distribution and performed worse than GPTmeans. This is explained by the adaptive power of lightweight models (GPTmeans has only eight data points) when much information is collected, and the predictive power of heavier models (GPTvoters contains 1′200 data points) when the downstream task has to be performed based on little collected information.

Another approach to combine the adaptive and predictive power of the pre-trained models was the replacement parameter $\gamma$. Instead of using less data for training, the synthetic data are continuously removed throughout the simulation. This, however, raises the challenge of setting the optimal point of full replacement. In Fig 7A, we compared different values of $\gamma$ and found that the optimal performance arises when full replacement is achieved at the break-even point. In that case, the pre-trained model performed better even for later users. This can be explained by the different queries of the models. Even though the number of queries is equal, the pre-trained model collected significantly different user-question pairs. Fig 7B shows that the overlap of queries remains below 71% after 1′000 users — even when the training data had been fully replaced after 50 users. This indicates that due to the initial training, more informative questions were selected that proved to be valuable for later users as well.

## Break-even points

To understand the occurrence of break-even points, we compared the performance of Cold-start and GPTvoters across different values of $K$ (see Fig 6). In all scenarios, the randomly initialised model met the predictive accuracy of the pre-trained model after $85 < N < 895$ users. We, therefore, also accept Hypothesis 2B, which states that break-even points exist where the initial advantage of the pre-trained models is eroded by real-world data. However, we found that the relation of $N$ and $K$ differs for both metrics and, therefore, depends on the downstream task.

For the downstream task of missing value imputation (measured by RMSE), break-even points come later the more answers users provide. When users provide more information ($K$), the model collects enough data after fewer users ($N$). There is a robust anti-proportionality of $K$ and $N$ given by $N * K = 4′500$. This means that — regardless of $N$ and $K$ specifically — when the number of incoming user interactions reaches 4′500, the randomly initialised model is sufficiently updated to reach the performance of the pre-trained model. This finding is useful in two ways: First, it quantifies the value of the GPTvoters training data; second, this number can be used to choose the hyperparameter $\gamma$ such that after 4′500 interactions, the training data will be fully replaced by incoming user interactions.

For the downstream task of candidate recommendations (measured in by CRA), the findings show two overlapping effects. Similar to the effect seen for missing value imputation, a high $K$ pulls the break-even point to smaller values of $N$. However, now there is a second effect that disturbs the anti-proportional relationship. As seen in Fig 6B, the overall performance of both models decreases if users provide fewer answers. We explain this with the difficulty of the task to identify the nearest neighbours from the candidates. If users provide fewer

answers (e.g., $K = 5$), it is impossible for any model to accurately estimate the user's characteristics, so the benefit of the training data is not evident. Therefore, the break-even point is already reached after 290 users, even though only $290 * 5 = 1'350 < 4'500$ user interactions were collected. If the users provide more answers ($K = 20$), the estimated characteristics of the users become more accurate, and the benefits of the initial training data become evident (resulting in a higher $N$). For many answers per user ($K > 30$), the learning of the randomly initialised model becomes faster. Thus, break-even points occur earlier again. Overall, we must, therefore, reject Hypothesis 2C, which states that there is a proportionality between the number of answers per user and the number of users before the break-even point. Instead, we find that this relationship depends on the downstream task.

## Limitations

While the proposed method to generate training data for the adaptive questionnaire with an LLM proved to work well, our approach has some limitations. Most importantly, our method requires an LLM knowledgeable in the target domain. We have seen that GPT-4 possesses this domain knowledge for answering political questionnaires in Switzerland. However, this might not be the case for all political systems worldwide. Furthermore, many applications where the cold start problem occurs (e.g., recommender systems) include preferences about movies or products. Here, LLMs might have difficulties simulating user interactions due to their inability to consume items. Hence, the generalization of the method might be limited to domains where the LLM can retrieve relevant information from the web.

Another limitation of our work concerns possible biases introduced by using LLM-generated training data for the question selection policy. While the synthetic data get replaced by real user interaction over time, there might be the risk of *path dependencies*. One possible scenario could be that due to its biased training data, the question selection policy will choose those questions for users that reinforce the initial bias. In our analysis, we investigated such effects by looking at the extremity of recommended candidates and did not find an increased bias compared to the cold start setting. However, there could be a more complex introduction of bias that this work did not investigate.

Furthermore, we recognise limitations in the overall setup of our adaptive questionnaire simulation. In some domains where adaptive questionnaires are used (such as education or healthcare), the setup might be different from ours. While educational testing usually uses one-dimensional latent spaces, adaptive questionnaires in healthcare are not evaluated by recommendation accuracy but feature selection [67]. These metrics were, due to our focus on the political domain, not included in our analysis. Furthermore, we exclusively focused on one particular question selection strategy, i.e., an uncertainty-based approach. In the context of recommender systems, other strategies have been proposed that specifically address the exploitation/exploration trade-off and path dependencies [68]. Including them in our simulation could, therefore, generalise the results.

Lastly, our simulation required an additional parameter to specify how fast real user interactions replace the training data. In our analysis, we developed simple heuristics on how to set this parameter a priori. However, the optimal value of $\gamma$ might be influenced by the quality of the LLMs predictions, the noise of the users' answers, and the difficulty of the downstream task. Future work can focus on choosing this parameter more systematically: analytically, where possible, or empirically by learning it.

## Conclusion

In this work, we explored the potential of LLM-generated datasets to pre-train the statistical model of an adaptive questionnaire in the absence of other training data. This addresses the cold start problem, which currently limits their application. Our study was divided into two parts: First, we evaluated how well GPT-4 could produce such a training dataset by comparing its generated interactions to real candidates' answers in a political questionnaire. Second, we measured the performance of the statistical model with and without this training data in two applications: wiki surveys and VAAs.

The results of the first experiment indicated that GPT-4 has high-quality domain knowledge of Swiss politics. The generated synthetic data points were within one standard deviation of the real candidates' answers of their respective parties for 85.3% of the questions. However, their overall distribution showed less variance and overfitted the party-means. To mitigate these shortcomings, we proposed a method to interpolate the generated samples, which future work could extend and validate with other datasets.

The results of the second experiment provided robust evidence that GPT-4 generated training data can reduce the cold start problem of adaptive questionnaires in political surveys. The statistical model with pre-training significantly outperformed the randomly initialised model for early users. The break-even point relied on the number of interactions each user provided. The relationship between the number of interactions per user and the break-even point depended on the downstream task. For the first task, missing value imputation, there was a clear negative correlation, i.e., the more answers per user, the earlier the break-even point. For the second task, candidate recommendations, no monotonous dependency could be found. This motivates future work to find ways to predict break-even points when using the method in practice.

In summary, this work proposed a cheap and versatile approach to train adaptive questionnaires. Wiki surveys in the political domain could especially benefit from the improved data collection method as they commonly contain too many questions for users to answer, and no prior training data exists to effectively select the most informative ones. The proposed framework demonstrated promising results, paving the way for effective data collection in political surveys.

## Supporting information

**S1 Text.** Additional Figures and Tables.
(PDF)

## Acknowledgments

We thank the team from *Politools* for providing the *Smartvote* data and David Camorani for his invaluable advice on scientific writing.

## Author contributions

**Conceptualization:** Fynn Bachmann, Daan van der Weijden, Cristina Sarasua, Abraham Bernstein.

**Data curation:** Fynn Bachmann.

**Formal analysis:** Fynn Bachmann.

**Funding acquisition:** Cristina Sarasua, Abraham Bernstein.

**Investigation:** Fynn Bachmann, Daan van der Weijden.

**Methodology:** Fynn Bachmann, Daan van der Weijden, Cristina Sarasua, Abraham Bernstein.

**Project administration:** Fynn Bachmann, Abraham Bernstein.

**Resources:** Abraham Bernstein.

**Software:** Fynn Bachmann, Daan van der Weijden.

**Supervision:** Cristina Sarasua, Abraham Bernstein.

**Validation:** Fynn Bachmann.

**Visualization:** Fynn Bachmann, Daan van der Weijden.

**Writing – original draft:** Fynn Bachmann, Daan van der Weijden, Lucien Heitz.

**Writing – review & editing:** Fynn Bachmann, Daan van der Weijden, Lucien Heitz, Cristina Sarasua, Abraham Bernstein.

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
