## [Decision Letter · Decision Letter 0]

24 Jan 2025

PONE-D-24-59142Adaptive questionnaires and GPT-4: Tackling the cold start problem with simulated user interactionsPLOS ONE

Dear Dr. Bachmann,

Thank you for submitting your manuscript to PLOS ONE. After careful consideration, we feel that it has merit but does not fully meet PLOS ONE’s publication criteria as it currently stands. Therefore, we invite you to submit a revised version of the manuscript that addresses the points raised during the review process.

We look forward to receiving your revised manuscript.

Kind regards,

Carlos Carrasco-Farré

Academic Editor

PLOS ONE

Journal Requirements:

“Support of the Swiss National Science Foundation (SNSF) under grant ID CRSII5-205975 provided the primary funding for our research. Additionally, this work was partially supported by the Digital Society Initiative (DSI) of the University of Zurich through a grant from the DSI Excellence Program.”

“We gratefully acknowledge the support of the Swiss National Science Foundation (SNSF) under grant ID CRSII5-205975, which provided the primary funding for our research. Additionally, this work was partially supported by the Digital Society Initiative (DSI) of the University of Zurich through a grant from the DSI Excellence Program. We also extend our thanks to the team from Politools for providing the Smartvote data, and to David Camorani for his invaluable advice on scientific writing.”

“Support of the Swiss National Science Foundation (SNSF) under grant ID CRSII5-205975 provided the primary funding for our research. Additionally, this work was partially supported by the Digital Society Initiative (DSI) of the University of Zurich through a grant from the DSI Excellence Program.”

Additional Editor Comments:

Dear Authors,

Thank you for submitting your manuscript to PLOS ONE, which addresses the use of Large Language Models (LLMs) to mitigate the cold start problem in adaptive questionnaires. The reviewers have provided valuable feedback, which highlights both the strengths of your work and areas requiring substantial revision. After careful consideration of their comments, I am recommending a major revision for your manuscript, with confidence that addressing the concerns raised will significantly improve the quality and impact of your work. Below, I summarize the main points raised by the reviewers, grouped into major and minor concerns, to guide your revision process. However, please, provide an answer for each of the comments raised by reviewers, regarding of my highlighting.

Major Concerns to Address

1. Experimental Design and Evaluation Metrics

Reviewer 1 emphasized the need for a clearer rationale for selecting evaluation metrics such as RMSE and Candidate Recommendation Accuracy (CRA). Additionally, the inclusion of complementary metrics (e.g., groundedness or faithfulness scores) is suggested to provide a more comprehensive assessment of the system’s performance.

2. Generalizability Beyond the Political Domain

Your experiments focus exclusively on the Swiss political domain. Reviewer 1 recommends either conducting experiments in other domains (e.g., education or healthcare) or providing an in-depth discussion about potential challenges and adaptations when applying your approach to other fields.

3. Synthetic Data Diversity and Limitations

Reviewer 1 noted the limitations of GPT-4-generated data, particularly its tendency to lack diversity and perform poorly in edge cases. A more quantitative analysis of how these limitations affect model performance—especially in non-typical scenarios—is needed.

4. Decay Parameter (γ) Analysis

Reviewer 1 highlighted the insufficient analysis of the decay parameter (γ). While you mention that a sensitivity analysis will be presented in a future paper, some discussion and practical guidance for parameter tuning in real-world applications are needed in the current work.

5. Statistical Testing of Data Distributions

Reviewer 2 pointed out that while the downstream performance of GPT-generated data is presented as the gold standard, statistical tests distinguishing the distributions of synthetic and real data should be included. This would strengthen your claim that LLM-generated data approximates real-world distributions.

6. Consistency in Experimental Setup

Reviewer 2 questioned inconsistencies in user dropout points in Fig. 4A and Fig. 4B. Clarifying the rationale for these differences and aligning the setups, if possible, would enhance the paper's coherence.

Minor Concerns and Suggested Improvements

Line 276: Reviewer 2 suggests adding a plot showing decision boundaries for all questions on the same PCA plot to confirm alignment with principal dimensions.

Line 277: Provide a rationale for binarizing Likert scores or explore alternatives, such as using normalized scores directly.

Line 329: Clarify discrepancies in values in Table 3 (e.g., 0.165 vs. 0.191).

Line 334: Explain the source of missing values.

Figures: Address issues such as untranslated legends (e.g., Fig. 7) and missing data points (e.g., Fig. 7 discussion). Ensure clarity and consistency across all figures.

Party Labels (Line 210): For readers unfamiliar with Swiss politics, label parties as left/right and liberal/conservative in relevant figures and discussions.

Typographical Errors: Address minor errors, such as missing words (e.g., “accuracy” in Line 484) and incorrect terms (e.g., "LMM" instead of "LLM" in Line 512).

Editorial Decision and Next Steps

The reviewers have recognized the potential and novelty of your work while also identifying significant gaps that need to be addressed. I encourage you to revise the manuscript thoroughly, incorporating the feedback from both reviewers. A systematic response to each comment will be essential to demonstrate how their concerns have been addressed.

Once the revised manuscript is submitted, it will be re-evaluated to ensure that the revisions adequately address the reviewers' concerns. We believe that with these changes, your manuscript will make a meaningful contribution to the field and advance the understanding of LLM applications in adaptive questionnaires.

We look forward to receiving your revised manuscript. Should you have any questions about the reviewers' feedback or the revision process, please do not hesitate to contact us.

Best regards,

Reviewers' comments:

Reviewer's Responses to Questions

**Comments to the Author**

1. Is the manuscript technically sound, and do the data support the conclusions?

Reviewer #1: Yes

Reviewer #2: Yes

2. Has the statistical analysis been performed appropriately and rigorously? 

Reviewer #1: Yes

Reviewer #2: Yes

3. Have the authors made all data underlying the findings in their manuscript fully available?

Reviewer #1: Yes

Reviewer #2: Yes

4. Is the manuscript presented in an intelligible fashion and written in standard English?

Reviewer #1: Yes

Reviewer #2: Yes

5. Review Comments to the Author

Reviewer #1: The draft presents an alternative approach to mitigate the cold start problem in adaptive questionnaires using Large Language Models (LLMs). The work is methodologically interesting and indicates some potential of LLMs to generate training data. However, I suggest a major revision to address certain issues and enhance the overall quality of the research contribution. Below, I outline four specific areas for improvement:

The experimental design of the study needs further elaboration, particularly regarding the evaluation metrics. While Root Mean Squared Error (RMSE) and Candidate Recommendation Accuracy (CRA) are utilized effectively, the paper does not justify why these metrics were chosen over others, such as Groundedness Score or faithfulness (a measure to evaluate hallucination probability). This lack of discussion leaves the reader unclear on whether the selected metrics fully capture the performance of the adaptive questionnaire in all its dimensions. To improve the rigor of the evaluation, the authors should provide a clear rationale for their choice of metrics and consider including complementary measures that might give a more holistic view of the system's performance.

Another aspect of the experimental design that requires attention is the generalizability of the proposed method. The paper's experiments focus exclusively on the Swiss political domain, which raises questions about whether the findings can be applied to other areas where adaptive questionnaires are used, such as education or healthcare. The lack of broader validation limits the applicability of the study's conclusions. It would strengthen the paper by conducting additional experiments in diverse domains or providing a detailed discussion about the potential challenges and adaptations required for applying the method beyond political surveys.

The paper also highlights a limitation in the synthetic data generated by GPT-4: it tends to be overly centered and lacks the diversity seen in real-world data, potentially leading to overfitting or poor performance in edge cases. However, this limitation has not been thoroughly analyzed. The authors could provide a more quantitative investigation into how this affects the performance of their models, particularly in scenarios with outlier user profiles or other non-typical cases. Such an analysis would add depth to the discussion and demonstrate a complete understanding of the method's limitations.

Finally, the paper introduces a decay parameter, γ, for the GPTreplace dataset, which determines how quickly the synthetic data is replaced by real user interactions. While this parameter is important for the system's performance, its impact is not systematically analyzed, and the paper does not offer clear guidance on how it should be set in practice. Even authors claim that a complete sensitivity analysis of this parameter will be done in a separate paper, adding some discussion on practical tuning strategies, would greatly enhance the utility of the study for practitioners seeking to apply the method in real-world scenarios.

By addressing these points, the authors can improve the robustness, clarity, and applicability of their work, ensuring it has a broader impact in the field.

Reviewer #2: This paper addresses the cold start problem in adaptive questionnaires by using GPT-4 to generate synthetic training data, demonstrating how LLMs can effectively simulate political survey responses to improve early-stage questionnaire performance. Strengths include clear hypotheses, good experimental design with various GPT-generated datasets, two downstream tasks evaluated, and detailed discussion of results and limitations.

Main concerns:

1. Line 276: A plot where the decision boundaries for all questions shown on the same plot would be interesting. The reason is that if most boundaries align with the first or the second principal dimension, then we know that both the LR model and the choice of using PCA with 2 components are making sense.

2. Line 277: Is binarization necessary? Can you directly use the normalized Likert score as a feature? What's the rationale behind this?

3. Line 329: These numbers are 0.165 and 0.191 respectively in Table 3. Please clarify

4. Line 334: What is causing the missing values?

5. Fig 4: Fig 4 does show that the cold start problem exists and the proposed method is able to alleviate it. Once question though is why in Fig 4A users answer 10 questions before dropout while in Fig 4B users answer 40 questions before dropout. Would it be better to keep this consistent? Curious about the rationale about this.

6. Line 490: The discussion on the observed trend for break-even point for CRA could be improved. I would say that when K is very small (<20), there is no way to accurately estimate each user's characteristics, hence the benefit of the initial training data is not evident. My understanding is not necessarily correct but I think is more clear.

7. Statistical tests for distinguishing distributions: This paper includes several demonstrations of how GPT generated data is close to the real data distribution, but no statistical tests are performed. Although the downstream performance is the gold standard, at least we can know if statistical tests are useful at all in gauging the quality of LLM-generated synthetic data.

Minor issues

1. Line 194: exactly how many candidates and voters are there?

2. Line 210: for people unfamiliar with Swiss parties, can you label which parties are left/right and liberal/conservative? You might also want to label the axes in the figure.

3. Line 349: SP also lies outside the Gaussian fit

4. Line 484: missing the word “accuracy”

5. Line 512: LMM -> LLM

6. In some of the figures, the legend is Swedish. Consider replace them with English, or make sure that the translation is provided in the caption or related discussion in the main text. Otherwise it causes confusions and guessing, e.g. in Fig. 7 the Green party.

7. Fig 7: what is the standard deviation of GPT4 generated answers? The std of the candidates' answers show interesting patterns across parties. For example, for Mitte, the std is the largest for questions with the most balanced answers, as indicated by the convex shape of the silhouette of all std bars. Another example is Green, where the concave shape shows that the questions with extreme answers have the most variance. It would be interesting to see if GPT4 can also mimic this, and thus further justify the use of LLMs.

8. Fig 7 discussion: For question 32253, the black circle seems to be missing. And for question 32238, there seems to be no circle pointing it. So I’m not sure where to find it in Fig 7.

6. PLOS authors have the option to publish the peer review history of their article (what does this mean?). If published, this will include your full peer review and any attached files.

Reviewer #1: No

Reviewer #2: No

---

## [Author Response · Author response to Decision Letter 1]

12 Mar 2025

Detailed response to the comments of Reviewer 1

We numbered the comments of the reviewer according to their order and added our responses below each comment. When we mention line numbers, we refer to the marked-up copy of the revised manuscript with highlighted changes. When we mention figures, we refer to their enumeration in the revised manuscript (in-stead of the figure enumeration of the initial submission). We would like to thank the reviewer for their time and insightful comments.

Comment of Reviewer #1.1

The experimental design of the study needs further elaboration, particularly regarding the evaluation metrics. While Root Mean Squared Error (RMSE) and Candidate Recommendation Accuracy (CRA) are utilized effectively, the paper does not justify why these metrics were chosen over others, such as Groundedness Score or faithfulness (a measure to evaluate hallucination probability). This lack of discussion leaves the reader unclear on whether the selected metrics fully capture the performance of the adaptive questionnaire in all its dimensions. To improve the rigor of the evaluation, the authors should provide a clear rationale for their choice of metrics and consider including complementary measures that might give a more holistic view of the system's performance.

Our response to #1.1:

We clarified our use of metrics in the revised manuscript (Methods section, line 317-342). We use RMSE/CRA to evaluate the downstream task after the data collection (to compare the question selection and prediction quality of the statistical model). We believe that these metrics fully capture the performance of each down-stream task (RMSE for missing value imputation is well established, and CRA as the overlap of recommended candidates with the ground truth candidates is intuitive for candidate recommendation).

The reviewer suggests adding Groundedness score or faithfulness score. However, both these metrics evaluate LLM generated answers relative to some document, which corresponds to another task. To evaluate our LLM generated answers, we compare them to real candidates. Groundedness or Faithfulness score do not necessarily fit here because the LLM is not generating text, but only a number. This numerical response is not easy to check for ‘faithfulness’ other than comparing it to the number provided by the real candidates, which is already being done.

Comment of Reviewer #1.2

Another aspect of the experimental design that requires attention is the generalizability of the proposed method. The paper's experiments focus exclusively on the Swiss political domain, which raises questions about whether the findings can be applied to other areas where adaptive questionnaires are used, such as education or healthcare. The lack of broader validation limits the applicability of the study's conclusions. It would strengthen the paper by conducting addi-tional experiments in diverse domains or providing a detailed discussion about the potential challenges and adaptations required for applying the method beyond political surveys.

Our response to #1.2:

We changed the title of the paper from “Adaptive questionnaires and GPT-4: Tackling the cold start problem with simulated user interactions” to “Adaptive political surveys and GPT-4: Tackling the cold start problem with simulated user interactions”.

In our analysis, we exclusively focus on the Swiss political domain. We believe that this focus is general enough to argue that the method can be applicable to political questionnaires in many (Western) democracies. Add-ing another political dataset would not add much evidence to this claim. Testing the method for a domain out-side political questionnaires (such as healthcare or education) would exceed the scope of this paper. We acknowledge this limitation by narrowing the focus of the paper with a more precise title. Additionally, we added more details to the Limitations section in line 658-663 and to the Conclusion section in line 701-709.

Comment of Reviewer #1.3

The paper also highlights a limitation in the synthetic data generated by GPT-4: it tends to be overly centered and lacks the diversity seen in real-world data, potentially leading to overfitting or poor performance in edge cases. However, this limita-tion has not been thoroughly analyzed. The authors could provide a more quantitative investigation into how this affects the performance of their models, particularly in scenarios with outlier user profiles or other non-typical cases. Such an analysis would add depth to the discussion and demonstrate a complete understanding of the method's limitations.

Our response to #1.3:

In our initial submission, we showed a lack of diversity in the synthetic data by comparing the GPT-4 samples to real candidates and concluded that they overfit to the party-mean. We then addressed the consequences of that overfitting for the adaptive questionnaire simulation in two ways:

- We created the GPTvoters dataset as interpolations to obtain a more homogeneous dataset.

- We used the replacement parameter to remove parts of the training data and replace them with in-coming user interactions.

However, we agree with the reviewer that we did not investigate the effects of the lack of diversity in the syn-thetic data in “scenarios with outlier user profiles or other non-typical cases”. To address this concern, we now provide a more quantitative investigation into how overfitting affects the performance of our models. Consid-ering “non-typical cases” voters at the political extremes (measured by their distance to centre of the embed-ding) and candidates with overlapping party positions (like SP and Greens). The exact methodology and re-sults are provided in the Methods section (line 205-207), the Results section (line 479-493), and in the appendix in Fig 15 and Table 5. We then discussed the results in the Discussion section (line 526-538) and Limitations section (line 653-656).

Comment of Reviewer #1.4

Finally, the paper introduces a decay parameter, �, for the GPTreplace dataset, which determines how quickly the synthetic data is replaced by real user interactions. While this parameter is important for the system's performance, its impact is not systematically analyzed, and the paper does not offer clear guidance on how it should be set in practice. Even authors claim that a complete sensitivity analysis of this parameter will be done in a separate paper, adding some discussion on practical tuning strategies, would greatly enhance the utility of the study for practitioners seeking to apply the method in real-world scenarios.

Our response to #1.4:

This is a very good point. For our revised manuscript, we conducted a systematic analysis of this replacement parameter. To this end, we slightly changed the setup of our experiment:

1. Instead of considering the replacement parameter as part of a separate training dataset (GPTre-place), we view it as a property of the adaptive questionnaire, i.e., the statistical model, itself. This is explained in line 309-315.

2. We removed the GPTreplace dataset from Fig 5 (as GPTreplace no longer exist as a dataset). This is explained in the Methods section (line 238 and line 260-264).

3. We simulate the adaptive questionnaire for different values of � and present the results in an addi-tional paragraph in the Results section (line 495-508).

4. We add another figure with the results for the analysis (Fig 7) and discuss the findings in the Discussion section (line 582-600).

5. We reformulate the Limitations section to address this new setup (line 668-674).

Previously, the GPTreplace dataset was a combination of training dataset and simulation parameter; now these two aspects are clearly separated in the Methods, Results, Discussion, and Limitations section. While do not provide clear guidance on how to set the replacement parameter in practice, we offer additional data for understanding and deliver heuristics for practical applications. 

Detailed response to the comments of Reviewer 2

We numbered the comments of the reviewer according to their order and added our responses below each comment. When we mention line numbers, we refer to the marked-up copy of the revised manuscript with highlighted changes. When we mention figures, we refer to their enumeration in the revised manuscript (in-stead of the figure enumeration of the initial submission). We would like to thank the reviewer for their time and insightful comments.

Comment of Reviewer #2.1

Line 276: A plot where the decision boundaries for all questions shown on the same plot would be interesting. The reason is that if most boundaries align with the first or the second principal dimension, then we know that both the LR model and the choice of using PCA with 2 components are making sense.

Our response to #2.1:

We agree with this commend of the reviewer. We provide such a plot in Fig 12 in the appendix and refer to it in the Methods section (line 281-284). Most decision boundaries align with the first principal component.

Comment of Reviewer #2.2

Line 277: Is binarization necessary? Can you directly use the normalized Likert score as a feature? What's the rationale behind this.

Our response to #2.2:

We clarified this in line 279. The reason why we binarize the normalized Likert scores, is that our statistical model (Logistic Regression) does not accept continuous variables as class labels. This also holds for most item-response theory models that are commonly used in political science (see Related work section). There might be alternatives, but we didn’t aim to optimize the statistical model in this experiment. Instead, our goal was to measure the effect of the training data while keeping the model fixed.

Comment of Reviewer #2.3

Line 329: These numbers are 0.165 and 0.191 respectively in Table 3. Please clarify.

Our response to #2.3:

We restructured our Results section for clarification. We calculate two distances:

- GPT-4 Distance: The distance of one random GPT sample to its corresponding party-mean and the standard deviation of this estimator (Table 2)

- Candidate Distance: The distance of one random candidate to their corresponding party-mean and the standard deviation of this estimator (Table 2)

We take all samples for one party (GPT samples or candidates respectively) and compute their distance to the party line. This results in 75 distances per sample (one for each question). Then, we compute the RMSE of these distances per party to obtain the numbers in Table 2.

Our revised Results section covers these different analyses in different subsections to disentangle the results. Furthermore, we added more details in the text (line 351-362, line 377-380, and line 411-418).

Comment of Reviewer #2.4

Line 334: What is causing the missing values?

Our response to #2.4:

A missing value arises when there is no numeric value in the answer of GPT-4 (e.g. if it replies with “As an LLM, I don’t have a political standpoint …”). We explain this source of missing values in the Methods section “Prompt engineering” in line 227-228. For clarity, we added another explanation in the Results section (line 419-425) and in the caption of Table 4 in the appendix.

Comment of Reviewer #2.5

Fig 4 does show that the cold start problem exists and the proposed method is able to alleviate it. Once question though is why in Fig 4A users answer 10 questions before dropout while in Fig 4B users answer 40 questions before dropout. Would it be better to keep this consistent? Curious about the rationale about this.

Our response to #2.5:

We agree with reviewer. Fig 4A showed the results for K=10 and RMSE metric, while Fig 4B showed results for K=40 and CRA metric. The reason for this was to demonstrate the scope of the experiment, indicating that not only the metric was variable, but also K was varied from K=5 to K=45.

In our revised manuscript, Fig 5 (formerly Fig 4) now shows both panels with the results for K=30 . For all other K, we added the corresponding figures in the appendix (Fig 13 and Fig 14). We also clarified this in the Results section (line 440-441) and Discussion (line 575-581).

Comment of Reviewer #2.6

Line 490: The discussion on the observed trend for break-even point for CRA could be improved. I would say that when K is very small (<20), there is no way to accurately estimate each user's characteristics, hence the benefit of the initial training data is not evident. My understanding is not necessarily correct but I think is more clear.

Our response to #2.6:

The understanding of the reviewer is correct. We rephrased our discussion accordingly to improve the clarity in line 624-627.

Comment of Reviewer #2.7

Statistical tests for distinguishing distributions: This paper includes several demonstrations of how GPT generated data is close to the real data distribution, but no statistical tests are performed. Although the downstream performance is the gold standard, at least we can know if statistical tests are useful at all in gauging the quality of LLM-generated synthetic data.

Our response to #2.7:

In Table 2 (formerly Table 3), we perform statistical tests (i.e., the Welch’s t-test). We find that the distances of the GPT samples are significantly closer to the party-mean compared to the candidates’ distances. To empha-size this more prominently we define the party-means in the Methods section “Data” (line 204-205) and pre-sent more description of the analysis in the Results section (line 374-386).

Furthermore, we perform two additional analyses:

- We compare the means and the standard deviations of the synthetic and candidates’ data per ques-tion in Fig 4 (and Fig 9 in the appendix). Here, we compute the number of questions that GPT an-swered within the confidence interval of the candidates of the respective party and describe the re-sults in line 387-399.

- We provide confusion matrices of GPT samples and candidates in Fig 10 and Fig 11 in the appendix. Here, we answer the questions:

o How many GPT samples (per party) are closest to which party-mean? How do these numbers compare to the real distances between parties?

o How many candidates (per party) are closest to which GPT-mean? How do these numbers compare to the real distribution, i.e.: How many candidates (per party) are closest to their own party-mean?

Comment of Reviewer #2.8

Line 194: exactly how many candidates and voters are there?

Our response to #2.8:

There are 1′029 candidates and 25′783 voters in the Smartvote dataset. We provide these numbers in the Methods section “Data” in line 193-195.

Comment of Reviewer #2.9

Line 210: for people unfamiliar with Swiss parties, can you label which parties are left/right and liberal/conservative? You might also want to label the axes in the figure.

Our response to #2.9:

We added labels to all axes of the political maps and also added the political orientation of each party when specifically mentioning them in the discussion or figures. Furthermore, we added a table in the appendix where we provide the full party names, their English translation, and their political orientation (Table 3). We refer to this table in the Methods section “Data” (line 198-199).

Comment of Reviewer #2.10

Line 349: SP also lies outside the Gaussian fit

Our response to #2.10:

This observation is correct. We present the results more coherently in our restructured Results section (line 370-373). In addition, we changed the confidence intervals in Fig 3B to the 1-� interval. This yields more par-ties to lie outside the ellipses but increases the interpretability of the figure. In any case, this figure is just for qualitative analysis, as it merely shows the 2D-projection onto the principal components. For statistical tests, we use the exact distances as described in Table 2.

Comment of Reviewer #2.11

Line 484: missing the word “accuracy”

Our response to #2.11:

We corrected the sentence in line 619.

Comment of Reviewer #2.12

Line 512: LMM -> LLM

Our response to #2.12:

We corrected the typo in line 648-649.

Comment of Reviewer #2.13

In some of the figures, the legend is Swedish. Consider replace them with English, or make sure that the translation is provided in the caption or related discussion in the main text. Otherwise it causes confusi

---

## [Decision Letter · Decision Letter 1]

26 Mar 2025

Adaptive political surveys and GPT-4: Tackling the cold start problem with simulated user interactions

PONE-D-24-59142R1

Dear Dr. Bachmann,

Thank you for submitting your revised manuscript entitled "Adaptive political surveys and GPT-4: Tackling the cold start problem with simulated user interactions" to PLOS ONE.

We have now received the expert reviewers’ evaluations of your revised submission. I am pleased to inform you that all reviewer concerns have been thoroughly addressed, and your manuscript has been accepted for publication.

Congratulations on this achievement. We appreciate the effort you and your co-authors have invested in preparing and revising your work. Your contribution will be a valuable addition to the literature on adaptive survey methodologies and the use of large language models in political science.

Kind regards,

Carlos Carrasco-Farré

Academic Editor

PLOS ONE

Additional Editor Comments (optional):

Thank you for submitting your revised manuscript entitled "Adaptive political surveys and GPT-4: Tackling the cold start problem with simulated user interactions" to PLOS ONE.

We have now received the expert reviewers’ evaluations of your revised submission. I am pleased to inform you that all reviewer concerns have been thoroughly addressed, and your manuscript has been accepted for publication.

Congratulations on this achievement. We appreciate the effort you and your co-authors have invested in preparing and revising your work. Your contribution will be a valuable addition to the literature on adaptive survey methodologies and the use of large language models in political science.

Should you have any questions about the next steps, including production or publication timelines, please don’t hesitate to contact us.

Reviewers' comments:

Reviewer's Responses to Questions

**Comments to the Author**

1. If the authors have adequately addressed your comments raised in a previous round of review and you feel that this manuscript is now acceptable for publication, you may indicate that here to bypass the “Comments to the Author” section, enter your conflict of interest statement in the “Confidential to Editor” section, and submit your "Accept" recommendation.

Reviewer #1: All comments have been addressed

Reviewer #2: All comments have been addressed

2. Is the manuscript technically sound, and do the data support the conclusions?

Reviewer #1: Yes

Reviewer #2: Yes

3. Has the statistical analysis been performed appropriately and rigorously? 

Reviewer #1: Yes

Reviewer #2: Yes

4. Have the authors made all data underlying the findings in their manuscript fully available?

Reviewer #1: Yes

Reviewer #2: Yes

5. Is the manuscript presented in an intelligible fashion and written in standard English?

Reviewer #1: Yes

Reviewer #2: Yes

6. Review Comments to the Author

Reviewer #1: All my comments have been addressed by the authors in this second version of the manuscript.

Reviewer #2: (No Response)

7. PLOS authors have the option to publish the peer review history of their article (what does this mean?). If published, this will include your full peer review and any attached files.

Reviewer #1: No

Reviewer #2: No

---

## [Editor Report · Acceptance letter]

PONE-D-24-59142R1

PLOS ONE

Dear Dr. Bachmann,

I'm pleased to inform you that your manuscript has been deemed suitable for publication in PLOS ONE. Congratulations! Your manuscript is now being handed over to our production team.

Kind regards,

on behalf of

Dr. Carlos Carrasco-Farré

Academic Editor

PLOS ONE